# A Cadaveric Study Using Anatomical Cross-Section and Computed Tomography for the Coelomic Cavity in Juvenile Cory’s Shearwater (Aves, *Procellariidae*, *Calonectris borealis*)

**DOI:** 10.3390/ani14060858

**Published:** 2024-03-11

**Authors:** Alejandro Morales Espino, Soraya Déniz, Marcos Fumero-Hernández, Mario Encinoso, Pascual Calabuig, Magnolia Conde-Felipe, José Raduan Jaber

**Affiliations:** 1IVC Evidensia Los Tarahales, 35013 Las Palmas, Spain; alejandro.morales108@alu.ulpgc.es; 2Hospital Veterinario, Facultad de Veterinaria, Universidad de Las Palmas de Gran Canaria, Trasmontaña, Arucas, 35413 Las Palmas, Spain; soraya.deniz@ulpgc.es (S.D.); marcos.fumero101@alu.ulpgc.es (M.F.-H.); 3Tafira Wildlife Rehabilitation Center (Cabildo de Gran Canaria), 35017 Las Palmas, Spain; pascual.calabuig@telefonica.net; 4Department of Pathology and Food Technology, Faculty of Veterinary Medicine, Universidad de Las Palmas de Gran Canaria, 35413 Las Palmas, Spain; magnolia.conde@ulpgc.es; 5Department of Morphology, Faculty of Veterinary Medicine, University of Las Palmas de Gran Canaria, Trasmontaña, Arucas, 35413 Las Palmas, Spain

**Keywords:** computed tomography, coelomic cavity, anatomical section, seabirds, Cory’s Shearwater

## Abstract

**Simple Summary:**

Cory’s Shearwater (*Calonectris borealis*) is a fascinating subject to explore the life and complexities of marine birds. Hence, we believe that knowledge of its coelomic cavity could provide essential information for biologists, veterinarians, and researchers. In this paper, we describe its coelomic cavity using anatomical cross-sections combined with computed tomography images to better understand the complex organization that encloses the different components of this cavity.

**Abstract:**

Birds play a crucial role in ecosystems, engaging in key functions such as pollination, pest control, and seed dispersal. The anatomical understanding of these species is essential to addressing emerging challenges, including climate change and habitat loss, which directly impact their survival. Detailed knowledge of avian anatomy is fundamental for research across various disciplines, ranging from ornithology to veterinary medicine. Therefore, this study aims to disseminate the understanding of avian anatomy and the application of computed tomography (CT) for visualizing the coelomic cavity in Cory’s Shearwater (*Calonectris borealis*). Recent advances in comprehending the anatomical structures of this region in avian species are highlighted, with a specific focus on Cory’s Shearwater as a study model. Various anatomical cross-sections and transverse CT images were described and analyzed in detail, offering a comprehensive insight into the coelomic cavity from different perspectives. The correlation between anatomical cross-sections and CT images is emphasized as crucial for a profound understanding of avian anatomy. This research contributes to the broader knowledge of avian anatomy, with potential implications for conservation efforts and veterinary practices.

## 1. Introduction

Cory’s Shearwater (*Calonectris borealis*), also known as the common shearwater, stands as a fascinating subject of study in the avian field, providing a unique window to explore the life and complexities of marine birds [1,2]. With a wingspan that can reach remarkable dimensions, its wings play a crucial role in its long migratory flights [1,3]. Its plumage, characterized by white and black tones, provides suitable camouflage in its marine environment. The aerodynamic shape of its body and its sharp beak are testaments to evolutionary adaptations that enable it to survive in the rigors of the oceanic environment and distinctive migratory habits. Cory’s Shearwater becomes an essential component of marine ecosystems, occupying vast areas ranging from the North Atlantic to the Indian and Pacific Oceans [1,2,3,4]. As a prominent member of the Procellariidae family, this species not only captures attention for its majestic presence but also for its vital role in the marine food chain [2,3]. Its specialized diet, focused on small fish, crustaceans, and cephalopods, positions it as a key component in the dynamics and balance of the oceans [1,2]. However, despite its ecological importance, Cory’s Shearwater faces significant challenges, from environmental threats to the consequences of human activity in its habitats [2,4]. As a consequence, this bird has most recently been assessed for the IUCN Red List of Threatened Species as having the least concern [5].

The anatomical investigation of birds holds fundamental importance in the scientific field, mainly for biologists, veterinarians, and researchers, providing a detailed and better understanding of the structure and function of the biological systems of these species [6,7,8]. In addition, the anatomical study is essential for elucidating the underlying mechanisms of avian physiology, and for comprehending the evolutionary adaptations that have allowed birds to inhabit a diversity of ecological niches [6,7,8].

Conventional radiology stands out as the predominant auxiliary diagnostic technique in avian medicine, primarily owing to its widespread availability in veterinary centers, cost-effectiveness, and non-invasive nature; nonetheless, contemporary imaging modalities, such as magnetic resonance (MR) and computed tomography (CT), offer advancements in avian diagnostic capabilities [9,10,11]. These modern techniques enable the acquisition of high-resolution three-dimensional images with minimal distortion and even allow for simulations of dynamic processes within different organs, vascular structures, and nerves, all achieved through a minimally invasive approach. A plethora of investigations exist concerning the application of diagnostic imaging to exotic species [12,13,14,15,16,17,18,19,20,21,22,23,24,25]. However, to our knowledge, a dearth of studies address the anatomical characteristics of the avian coelom utilizing Cory’s Shearwater as a model organism. Therefore, this study aimed to assess the coelomic cavity of Cory’s Shearwater through anatomical cross-sections and CT image examination, establishing it as a potential anatomical model for pathological investigations involving other phylogenetically related avian species.

## 2. Materials and Methods

### 2.1. Animals

To perform this investigation, 8 juvenile Cory’s (*Calonectris borealis*) cadavers were employed. The birds exhibited an average mass of 0.520 kg (ranging from 0.480 to 0.820 kg) and an average length, measured from the beak tip to the base of the tail, of 42 cm (ranging from 45 to 56 cm). The study group was provided by the Tafira Wildlife Rehabilitation Center (Consejeria de Área de Medio Ambiente, Clima, Energía y Conocimiento of the Cabildo de Gran Canaria, Spain) after a stranding event on the island coastline as a consequence of disorientation produced by light pollution. Six of our birds were quickly frozen for four days until the CT procedure, whereas the other two birds underwent dissection immediately after death to expose the coelomic cavity and visualize organ locations. This process facilitated the precise identification and correlation with the CT images.

### 2.2. CT Study

For the CT evaluation, we thawed our birds at room temperature for 12 h. We obtained transversal CT sequential images using a 16-slice helical CT scanner (Toshiba Astelion, Canon Medical System^®^, Tokyo, Japan). We positioned the birds symmetrically in dorsal recumbency on the stretcher with craniocaudal entry. We followed a standard protocol using the following parameters: 120 kVp, 80 mA, 512 × 512 acquisition matrix, 1809 × 858 field of view, a pitch of 0.94, and a gantry rotation of 1.5. The acquired images had a slice thickness of 0.6 mm. To elevate the identification of the different anatomical structures on CT, we used a variety of CT window settings by adjusting the window widths (WWs) and window levels (WLs): a bone window setting (WW = 1500; WL = 300), a soft tissue window setting (WW = 248; WL = 123), and a lung window setting (WW = 1400; WL = −500). We did not perceive substantial differences in CT density or anatomy in the coelomic cavity of the animals used in this study. Moreover, we used the original data to create volume-rendered reconstructed images using a standard dicom 3D format (OsiriX MD, Geneva, Switzerland).

### 2.3. Anatomical Sections

After the CT study, the carcasses underwent freezing at −80 °C for 72 h. Thereafter, we sectioned three birds, generating parallel sections one centimeter thick, using an electric band saw. The obtained sections were carefully irrigated with water to remove artifacts (feathers), which were removed using Adson forceps and identified before being photographed on both sides. 

### 2.4. Anatomic Evaluation

In pursuit of cross-sectional identification and labeling, concomitant with the associated CT images, we employed specific sources comprising textbooks and reports from scientific journals dedicated to bird anatomy [26,27,28,29]. Additionally, to achieve meticulous anatomical interpretation of coelomic structures, various anatomical preparations supplied by the Department of Morphology were utilized. These preparations were instrumental in augmenting our comprehension and precision of the organization of the coelomic cavity.

## 3. Results

A comprehensive array of illustrations delineating the anatomical intricacies of the Cory’s Shearwater coelomic cavity is depicted through a series of figures (Figure 1, Figure 2, Figure 3, Figure 4, Figure 5, Figure 6, Figure 7, Figure 8, Figure 9, Figure 10 and Figure 11). Figure 1 integrates diverse dissections, offering a comprehensive view of the principal formations included in the coelomic cavity. Figure 2 presents a multiplanar sagittal reconstruction (MPR) image, where the lines and their numerals (I–IX) denote an approximate level corresponding to subsequent anatomical and CT transverse planes. Figure 3, Figure 4, Figure 5, Figure 6, Figure 7, Figure 8, Figure 9, Figure 10 and Figure 11 exhibit a set of four images for each case: (A) an anatomical cross-section, (B) a bone CT window, (C) a soft tissue CT window, and (D) a pulmonary CT window. These figures are sequentially arranged in a rostrocaudal progression, commencing from the lungs in Figure 3 and concluding at the cloaca levels in Figure 11. Figure 12 comprises four images: (A) an anatomical dissection and (B–D) dorsal MPR volume rendering images showcased at varying levels. This series of figures collectively provides a detailed and nuanced representation of the anatomical features within the coelomic cavity of Cory’s Shearwater.

### 3.1. Anatomical Dissections and Cross-Sections

In this study, we provide anatomical dissections (Figure 1A–C and Figure 12A) and transversal cross-sections (shown in Figure 3A, Figure 4A, Figure 5A, Figure 6A, Figure 7A, Figure 8A, Figure 9A, Figure 10A and Figure 11A) of the coelomic cavity of the Cory’s Shearwater. These figures play a crucial role in aiding the recognition of organs belonging to the respiratory, circulatory, digestive, and urinary systems inside this cavity. This distinct anatomical feature in Cory’s Shearwater, observed through our meticulous examination, reflects the anatomical arrangement regarding the precise spatial disposition of vital organs within the coelomic cavity. Subsequently, we identified the heart that shows an oval conformation with a pointed apex located centrally inside this cavity and anterior to the liver (Figure 1A–C and Figure 12A). Furthermore, we observed the different components of the heart, encompassing the atria and the ventricles. Notably, the anatomical images played a pivotal role in visualizing remarkable vascular formations, including the brachiocephalic trunks, the left subclavian vein, the left and right jugular veins, and the left and right cranial cava veins (depicted in Figure 1A,B, Figure 3A, Figure 4A and Figure 12A, respectively). Moreover, both lungs were distinguishable, positioned in a craniodorsal location beneath the thoracic vertebrae and to the side of the ribs (Figure 1A,B and Figure 12A). The anatomical study also facilitated the identification of the trachea (Figure 1A,B and Figure 12A), coursing in a median position into the coelomic cavity before bifurcating into the right and left main bronchi. This bifurcation was clearly distinguished in the transverse anatomical sections (illustrated in Figure 3A and Figure 4A). In specific dissected images, we successfully identified the *syrinx* (Figure 1B, Figure 4A, and Figure 12A). Dorsal to the *trachea*, we identify additional structures, including the *esophagus*, the longissimus colli muscle, and the sternotracheal muscle (Figure 1A,B, Figure 3A, Figure 4A, Figure 5A, Figure 6A and Figure 12A, respectively). These images along the cross-section facilitated the observation of the different air sacs, including the cervical, thoracic, and abdominal air sacs (depicted in Figure 1A,B and Figure 4A, Figure 5A, Figure 6A and Figure 7A). These dissected images were helpful in distinguishing the thyroid gland (Figure 1B). It has an ovoid shape and is located at the base of the neck, just above the sternum bone. This gland consists of two lobes situated on either side of the trachea, connected by an isthmus.

Moreover, these anatomical dissections and sections also enabled the examination of the Cory’s Shearwater liver, comprising the right and left hepatic lobes (Figure 1A,C and Figure 12A). The organ was conspicuously large, and its cranioventral segments of both hepatic lobes enveloped the heart (as illustrated in Figure 1A–C, Figure 5A, Figure 6A and Figure 12A). Both lobes displayed comparable sizes, surpassing those of other organs within the coelomic cavity. The visceral surface of the organ is in close contact with the lungs. On the right, its visceral surface is in contact with a tubular structure that corresponds with the *duodenum* (Figure 7A). On the left visceral surface, it is associated with the glandular stomach (*proventriculus*) and with the muscular segment of the stomach referred to as the *ventriculus* (Figure 1C, Figure 7A, Figure 8A, Figure 9A and Figure 12A). Additionally, the medial border of the hepatic lobes exhibited the hepatopericardial ligament, establishing a connection between the liver and the heart apex (Figure 1A).

Additional elements of the digestive system, including the small and large bowel, were visible in these anatomical images. The dissected image located caudal to the liver provided a clearer depiction of the small intestine (*duodenum*) (Figure 1C and Figure 7A, Figure 8A, Figure 9A, Figure 10A, Figure 11A and Figure 12A). Within this region, the duodenum exhibited a distinct U-shaped configuration known as *ansa duodeni*, consisting of a descending segment, the *pars descendens*, and an ascending portion, the *pars ascendens* (Figure 1C and Figure 12A). In the area proximal to the stomach and situated between the duodenal loops and these two segments, we noted the pancreas within the *mesoduodenum*, exhibiting a pale-pink coloration (Figure 1C and Figure 12A). Further toward the caudal region, we were able to identify the terminal intestine (*caecum*) (Figure 1C and Figure 12A). The *caecum* appears well developed and is an elongated pouch connected to the junction of the small and large intestines. In addition, it exhibited a unique coloration, adding to its anatomical distinctiveness.

Moreover, the dissected images facilitated the identification of the spleen, showing a flattened rectangular shape and brown to cherry red color (Figure 1C and Figure 12A). It was medial to the proventriculus and ventriculus, close to the visceral surface of the liver. Furthermore, the employment of cross-sectional views proved indispensable in identifying both kidneys, which are positioned laterally to the spine and embedded dorsally within excavations of the synsacrum (Figure 1C, Figure 8A, Figure 9A, Figure 10A, Figure 11A and Figure 12A). A noteworthy aspect revealed by these sections is the close association of the ventral surfaces of the kidneys with the paired abdominal air sacs, providing insight into the intricate anatomical relationships. Additionally, the ureters, a vital conduit for transporting urine from the kidneys to the cloaca for excretion, were distinctly observed to originate from the ventral surface of the kidneys (Figure 1C and Figure 12A). In the posterior region of the coelomic cavity, we observed an excretory junction serving both the digestive and urogenital systems, referred to as the cloaca. This anatomical structure serves as a common passage for the digestive, urinary, and reproductive systems (illustrated in Figure 10A and Figure 11A). 

Furthermore, we observed various osseous structures pertaining to this cavity, encompassing the vertebrae, ribs, sternum, thoracic, and pelvic limbs, along with the pubis. These bony structures were intricately associated with different muscles, including distinct segments of the pectoral muscle (sternobrachialis, thoracic, and abdominal portions), as well as the supracoracoid, the scapulohumeralis, the scapulohumeral caudal, and the longissimus dorsi muscles (Figure 1A,B and Figure 3A, Figure 4A, Figure 5A, Figure 6A and Figure 7A).

### 3.2. Computed Tomography Images

For CT examination, we selected those images that meticulously aligned with their anatomical sections (Figure 3B,D, Figure 4B,D, Figure 5B,D, Figure 6B,D, Figure 7B,D, Figure 8B,D, Figure 9B,D, Figure 10B,D, Figure 11B,D and Figure 12B,C). Compared to the corresponding cross-section, these TC images offered enhanced morphological and tomographic insights into coelomic structures. Notably, CT images obtained with the pulmonary window setting (Figure 3B, Figure 4B, Figure 5B, Figure 6B, Figure 7B, Figure 8B, Figure 9B, Figure 10B, Figure 11B and Figure 12B,C) showed remarkable visualization of different bones, related muscles, and soft tissues. These results were similar to those obtained with the soft tissue and bone window settings. As a result, we differentiated between several skeletal formations, which included the ulna, thoracic vertebrae, ribs, sternum, humerus, femur, metacarpal bones, and phalanges. In addition, we examined a variety of muscles and ligaments connected to these skeletal elements, such as the scapulohumeralis, scapulohumeral caudal, longissimus colli, longissimus dorsi, pectoral, intercostal muscles, ventral collateral and cranial cubital ligaments, and fibularis longus muscles (Figure 3D, Figure 4D, Figure 5D, Figure 6D, Figure 7D, Figure 9D and Figure 12D). More precisely, this window facilitated the identification of the trachea and its bifurcation in the left and right major bronchi, the syrinx, the pulmonary vessels, and the honeycomb-like pattern of the lung parenchyma (Figure 3D, Figure 4D, Figure 5D and Figure 6D). This window also proved to be quite helpful in distinguishing the right and left primary bronchi entering into the ventromedial aspect of the lungs (Figure 4D). Other intrathoracic formations, including the heart and large vessels, such as the brachiocephalic trunks, were also identified (Figure 5D). Additionally, the walls of some air sacs, such as those of the cranial thoracic air sacs, could be distinguished (Figure 3D, Figure 5D, Figure 6D and Figure 7D). Regarding the digestive components, this particular CT window skillfully outlined the *esophagus*, the hepatic lobes, the glandular and muscular stomach components, various intestinal segments, and the cloaca (Figure 3D, Figure 5D, Figure 6D, Figure 7D, Figure 8D and Figure 9D). Additionally, the dorsal MPR volume rendering images distinguished the hepatopericardial ligament. This structure connects the liver to the pericardium, the membrane surrounding the heart (Figure 12B).

CT images acquired using the bone window and soft tissue settings (Figure 4B,C, Figure 5B,C, Figure 6B,C, Figure 7B,C, Figure 8B,C, Figure 9B,C, Figure 10B,C and Figure 11B,C) demonstrated a remarkable distinction between bones and soft tissues. Interestingly, the different heart chambers and main arteries could be distinguished with these windows (as seen in Figure 5B,C). Moreover, various digestive structures, such as the right and left hepatic lobes, different intestinal segments, as well as the glandular and muscular portions of the cory’s shearwater stomach, and the cloaca were also displayed (depicted in Figure 6B,C, Figure 7B,C, Figure 8B,C, Figure 9B,C, Figure 11B,C and Figure 12B,C). These windows were helpful in the visualization of the thoracic and abdominal air sacs, which were joined by connective tissue with adjacent organs or muscles (as shown in Figure 3B,C, Figure 4B,C, Figure 5B,C, Figure 6B,C and Figure 7B,C).

## 4. Discussion

In contrast to mammal species, birds do not present a structure separating the thoracic and abdominal cavities. As a result, these animals have a single cavity that contains all the viscera identified in mammals. To the best of the authors’ knowledge, no reports have become available concerning the anatomy of Cory’s Shearwater’s coelomic cavity using advanced imaging diagnostic techniques. Nonetheless, these techniques have been applied to other bird species routinely seen by specialists in the field, such as the domestic pigeon (*Columba livia domestica*), toco toucans (*Ramphastos toco*), and gray parrot (*Psittacus erithacus*), demonstrating that they are essential for enabling normal anatomy and abnormalities in the internal organs of birds [20,23,24]. Traditionally, clinicians have used imaging modalities such as radiography [30] and ultrasonography (usually mode B) [16,27], which have been essential in obtaining images of the bony and the architecture of the internal organs that compose this region. Modern imaging modalities, such as CT and MRI, have become essential techniques in veterinary medicine because they provide early identification of focal and diffuse changes. In addition, they allow views of the body from several tomographic planes without repositioning the animal, thereby displaying images with excellent anatomic resolution in the absence of tissue superimposition, high contrast between different structures, and exceptional tissue differentiation that allows the evaluation of spatial relationships between organs that are not easily detected using radiography or ultrasonography. Despite all these advantages, there are also notable drawbacks, including the excessive cost of this equipment, the relatively high cost of the procedure, the need to sedate and possibly restrain the bird, and the longer examination time in the case of MRI. In consequence, their use is valuable only in those cases where it is specifically indicated or for remarkable birds such as breeding and falconry birds [8,20]. Due to these shortcomings, traditional diagnostic imaging techniques are still the most routinely used because they are fast, low-cost, and widely available in daily avian practice. 

This investigation combined dissections and anatomical cross-sections, which showed essential information to help identify coelomic CT images using pulmonary, soft tissue, and bone window settings. Therefore, the utilization of dissections and anatomical cross-sections was immensely valuable in the depiction of the morphologic features of the Cory’s Shearwater coelomic cavity. Thus, the dissected images played a pivotal role in distinguishing the spleen and some components of the endocrine system, such as the thyroid gland located in the angle between the subclavian and common carotid arteries. The identification of this gland helped in visualizing other main vessels, such as the jugular veins adjacent to both glands and converging at the venous angle to form the right and left cranial cava veins, as well as the subclavian arteries. However, due to the wide interval between slices employed in this study, some structures were not clearly discernible in the transverse and dorsal CT images using different windows. Comparable investigations conducted in other exotic species, such as rabbits, guinea pigs, iguanas, loggerhead turtles, and sea birds, have consistently demonstrated that this combination is essential to anatomical studies based on regions or divisions and in visualizing the relations between various organs [11,15,17,20,23,31,32].

In this research, the CT images obtained using pulmonary, soft tissue, and bone window settings were pivotal in displaying the main structures constituting the coelomic cavity. Thus, the use of the pulmonary window provided notable advantages in the visualization of the respiratory components and the related vascular structures. Therefore, the course of the trachea together with the esophagus on the right side of the neck, the distinction of the syrinx at the tracheal bifurcation and the course of the two brachiocephalic trunks, which continue as the subclavian arteries, were displayed using this window setting. Additionally, this window was essential in distinguishing the left and right cranial cava veins. Similar results have been described in other CT studies conducted on neonatal foals [30], reptiles [8,16], and a variety of avian species [11,20], including toco toucans and Atlantic puffins. Despite the fact that we did not administer contrast media, the combination of the different windows contributed to the visualization of the heart chambers and their associated vascular structures. We assumed that it could be due to the larger size of the animal, which provided better resolution in contrast to other studies performed on smaller birds [11,20], where cardiac chambers were not defined on CT. As in other reports performed on sea birds [11], this CT-based morphological evaluation has demonstrated the suitability of cadavers for studying diverse anatomical patterns. However, it is important to highlight the absence of blood flow in dead animals when comparing the results with those obtained from live specimens using contrast media, especially in small birds [11,17,18]. 

This comprehensive imaging also facilitated an adequate examination of Cory’s Shearwater livers. Therefore, the precision afforded by the CT scans provided valuable insights about the liver. Thus, as observed in other avian species [18], the right lobe was larger than the left lobe. Moreover, the gallbladder was not only identified in the anatomical images but also in CT images. As explained before, it can be due to the larger size and volume of the bird, which could facilitate the visualization of these organs.

However, some limitations were observed, mainly due to work with fledglings that stranded as a consequence of artificial lights, which has been recognized as a relevant threat to biodiversity conservation. Birds can suffer from mass mortality episodes caused by artificial lights [33]. Thus, Cory’s Shearwater fledglings are attracted and disorientated by artificial lights when they are leaving their nests for the first time and fly towards the sea [34]. As a result, we could not identify the gonads in these animals, which could be explained because they were sexually immature animals. Nevertheless, several avian species consequently do not show phenotypic sexual dimorphism, making it necessary to use endoscopy or DNA testing for definitive sex determination [28]. 

## 5. Conclusions

This study elucidates the spatial organization of the different organs comprising the coelomic cavity of this bird by using different CT windows, anatomical cross-sections, and dissections. Our results have demonstrated the efficacy of CT imaging to assess the avian anatomical organization through pulmonary, soft tissue, and bone windows without using a contrast medium. These images could be used as an initial anatomical reference in Procellariidae birds. However, further studies on adult specimens should be performed to detect possible differences with adult birds.

## Figures and Tables

**Figure 1 animals-14-00858-f001:**
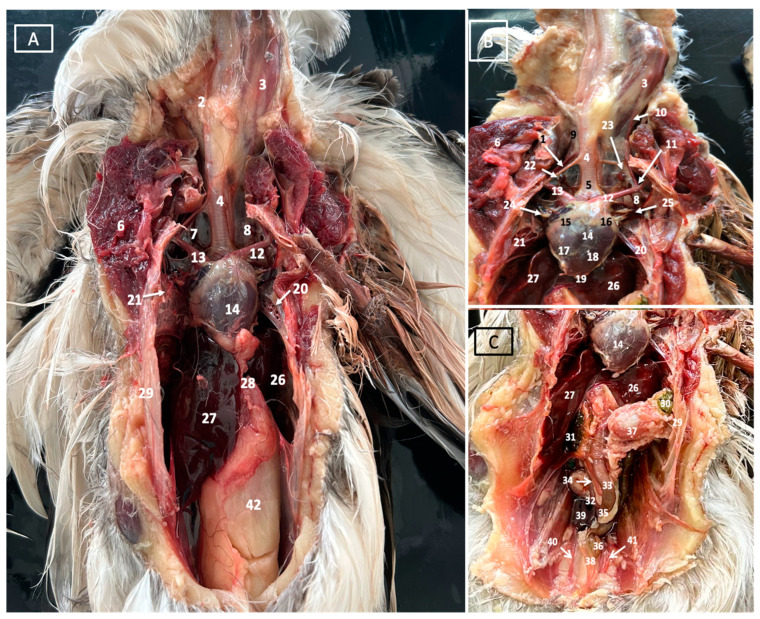
A gross dissection image illustrating the coelomic cavity of Cory’s Shearwater (**A**), along with detailed views of its cardiovascular (**B**) and digestive (**C**) structures. Anatomical labels include 1: sternotracheal muscle. 2: esophagus. 3: longissimus colli muscle. 4: trachea. 5: syrinx. 6: pectoral muscle (thoracobrachialis muscle). 7: right cervical air sac. 8: left cervical air sac. 9: right jugular vein. 10: left jugular vein. 11: left subclavian artery. 12: left brachiocephalic trunk. 13: right brachiocephalic trunk. 14: heart. 15: right atrium. 16: left atrium. 17: right ventricle. 18: left ventricle. 19: pericardium. 20: left lung. 21: right lung. 22: right thyroid. 23: left thyroid. 24: right cranial cava. 25: left cranial cava. 26: left hepatic lobe. 27: right hepatic lobe. 28: hepatopericardial ligament. 29: ribs: 30: ingesta: 31: spleen. 32: descending duodenum. 33: ascending duodenum. 34: pancreas. 35: duodenal-jejunal flexure. 36: caecum. 37: ventriculus. 38: jejunoileum. 39: right kidney. 40: right ureter. 41: left ureter. 42: intestinal peritoneal cavity.

**Figure 2 animals-14-00858-f002:**
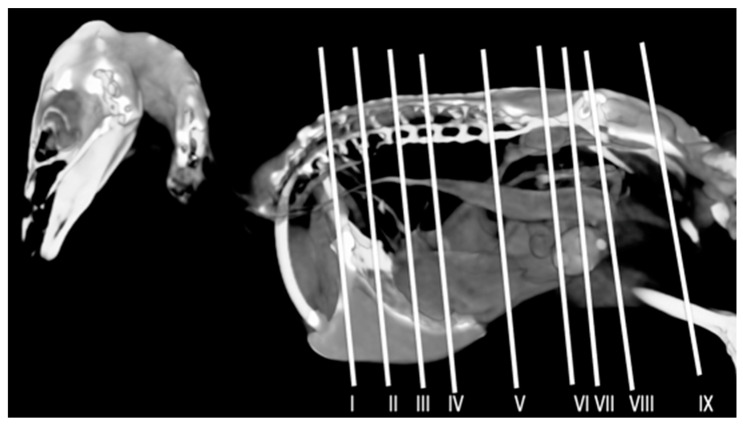
Sagittal MPR volume rendering image of a Cory’s Shearwater specimen. The lines and corresponding numbers (I–IX) indicate the approximate positions of the subsequent transversal cross-sections and CT images.

**Figure 3 animals-14-00858-f003:**
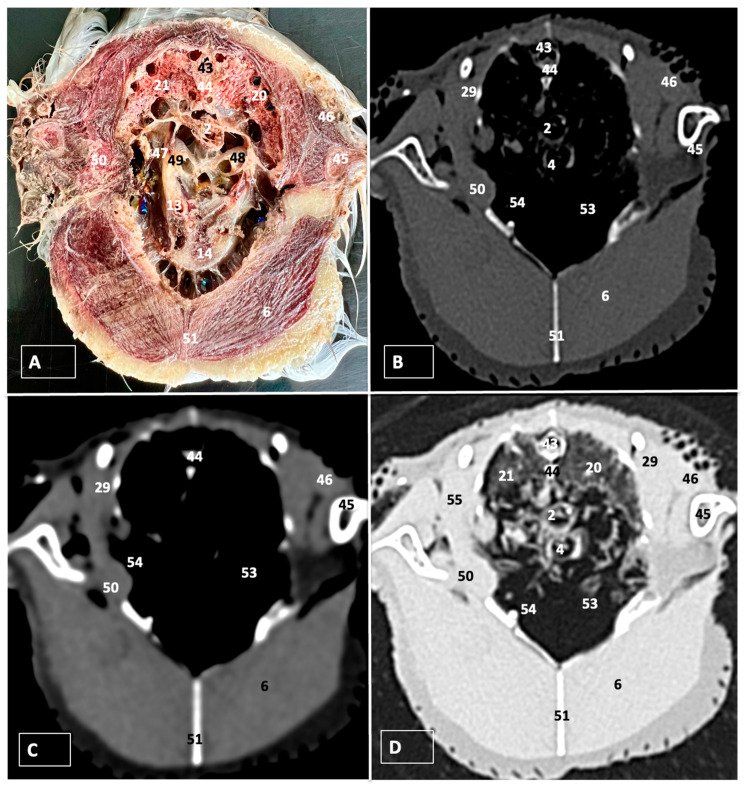
Transverse cross-section (**A**), bone (**B**), soft tissue (**C**), and pulmonary (**D**) CT images of the coelomic cavity of the Cory’s Shearwater at the level of the lungs corresponding to line I in Figure 2. 43: spinal cord. 44: vertebra (vertebral body). 21: right lung. 20: left lung. 2: esophagus. 4: trachea. 45: humerus. 46: scapulohumeralis muscle. 47: common carotid artery. 48: left primary bronchus. 49: right primary bronchus. 13: right brachiocephalic trunk. 14: heart. 46: scapulohumeral caudal muscle. 6: pectoral muscle (thoracobrachialis muscle). 51: sternum. 29: ribs. 53: cranial thoracic air sac (left). 54: cranial thoracic air sac (right). 55: intercostal muscle. 50: scapulohumeral caudal muscle.

**Figure 4 animals-14-00858-f004:**
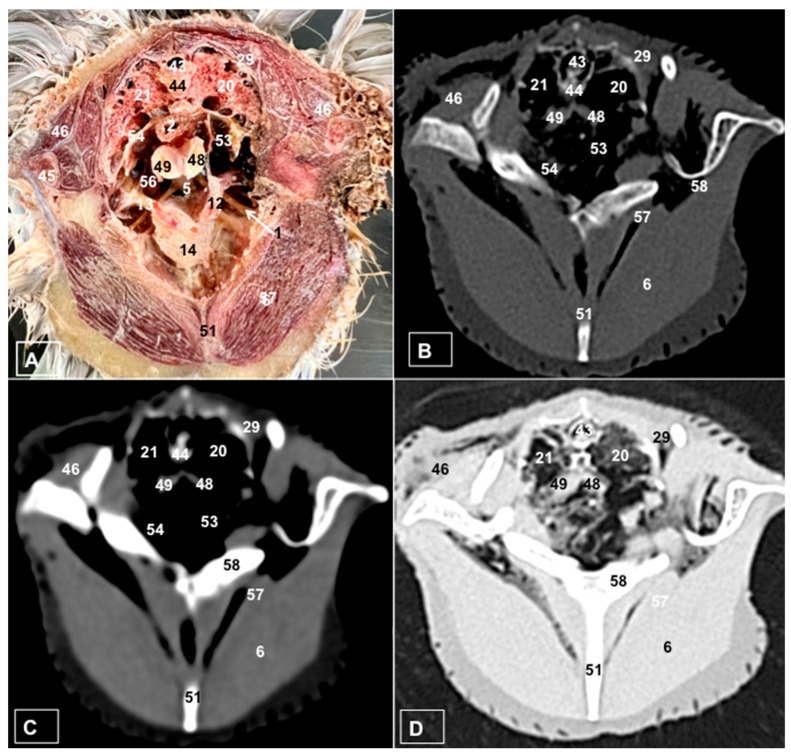
Transverse cross-section (**A**), bone (**B**), soft tissue (**C**), and pulmonary (**D**) CT images of the coelomic cavity of the Cory’s Shearwater at the level of the main bronchi corresponding to line II in Figure 2. 43: spinal cord. 44: vertebra (vertebral body). 29: rib. 21: right lung. 20: left lung. 46: scapulohumeralis muscle. 45: humerus. 2: esophagus. 54: cranial thoracic air sac (right). 53: cranial thoracic air sac (left). 5: syrinx. 14: heart. 13: right brachiocephalic trunk. 12: left brachiocephalic trunk. 56: right subclavian artery. 1: sternotracheal muscle. 48: left primary bronchus. 49: right primary bronchus. 6: pectoral muscle (thoracobrachialis muscle). 57: supracoracoid muscle. 51: sternum. 58: coracoid bone.

**Figure 5 animals-14-00858-f005:**
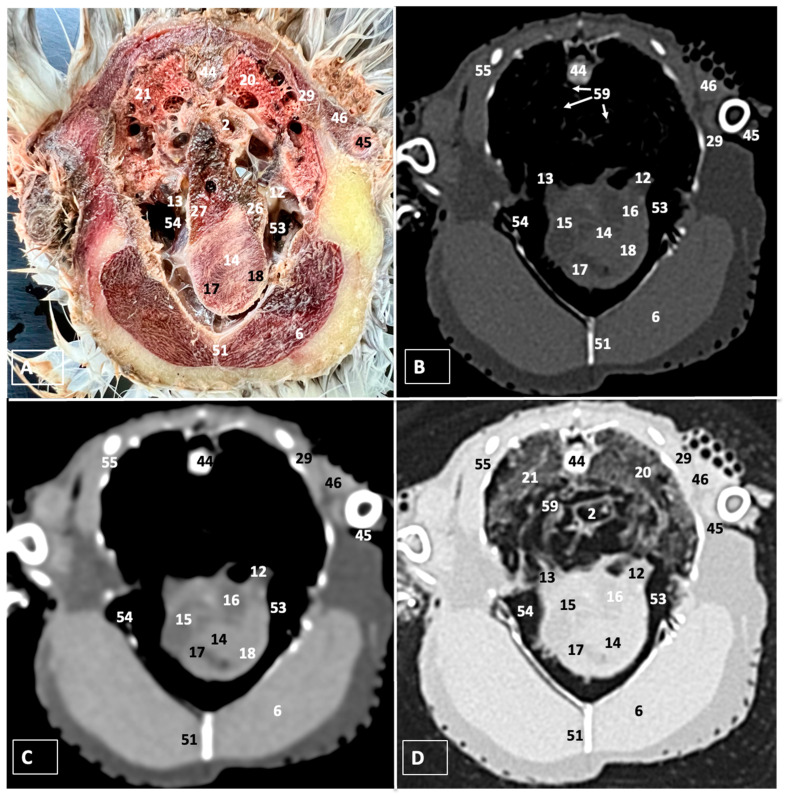
Transverse cross-section (**A**), bone (**B**), soft tissue (**C**), and pulmonary (**D**) CT images of the coelomic cavity of the Cory’s Shearwater at the level of the heart, corresponding to line III in Figure 2. 44: vertebra (vertebral body). 21: right lung. 20: left lung. 2: esophagus. 29: ribs. 46: scapulohumeralis muscle. 45: humerus. 54: cranial thoracic air sac (right). 53: cranial thoracic air sac (left). 26: left hepatic lobe. 27: right hepatic lobe. 14: heart. 17: right ventricle. 18: left ventricle. 15: right atrium. 16: left atrium. 13: right brachiocephalic trunk. 12: left brachiocephalic trunk. 59: pulmonary vessels. 51: sternum. 6: pectoral muscle (thoracobrachialis muscle). 55: intercostal muscle.

**Figure 6 animals-14-00858-f006:**
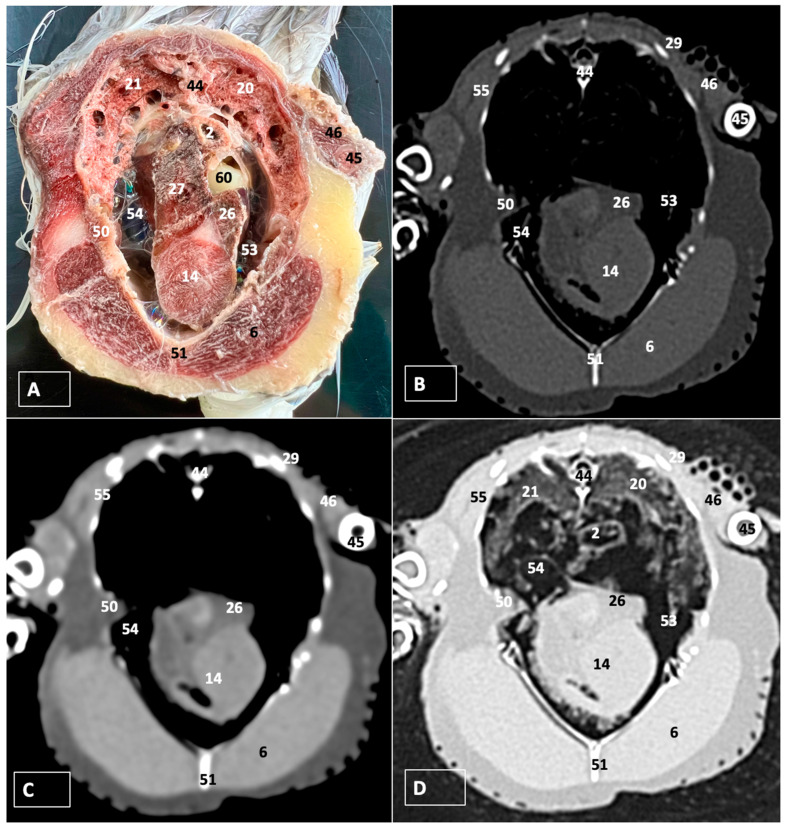
Transverse cross-section (**A**), bone (**B**), soft tissue (**C**), and pulmonary (**D**) CT images of the coelomic cavity of the Cory’s Shearwater at the level of the hepatic lobes corresponding to line IV in Figure 2. 44: vertebra (vertebral body). 2: esophagus. 46: scapulohumeralis muscle. 45: humerus. 21: right lung. 20: left lung. 50: scapulohumeral caudal muscle. 27: right hepatic lobe. 26: left hepatic lobe. 60: proventriculus. 53: cranial thoracic air sac (left). 54: cranial thoracic air sac (right). 14: heart. 51: sternum. 6: pectoral muscle (thoracobrachialis muscle). 55: intercostal muscle.

**Figure 7 animals-14-00858-f007:**
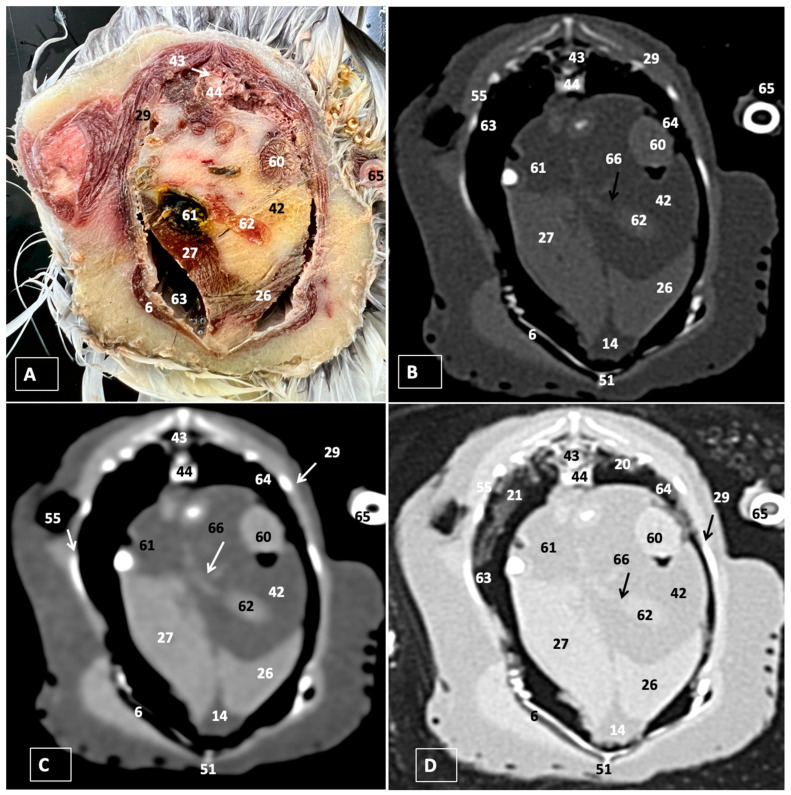
Transverse section (**A**), bone (**B**), soft tissue (**C**), and pulmonary (**D**) CT images of the coelomic cavity of the Cory’s Shearwater at the level of the caudal thoracic air sac corresponding to line V in Figure 2. 44: spinal cord. 43: vertebra (vertebral body). 29: ribs. 65: femur. 21: right lung. 20: left lung. 27: right hepatic lobe. 66: common hepatoenteric duct. 26: left hepatic lobe. 14: heart. 61: gallbladder. 42: intestinal peritoneal cavity (full of fat). 63: caudal thoracic air sac (right). 64: caudal thoracic air sac (left). 60: proventriculus. 62: duodenum. 51: sternum. 6: pectoral muscle (abdominal portion). 55: intercostal muscle.

**Figure 8 animals-14-00858-f008:**
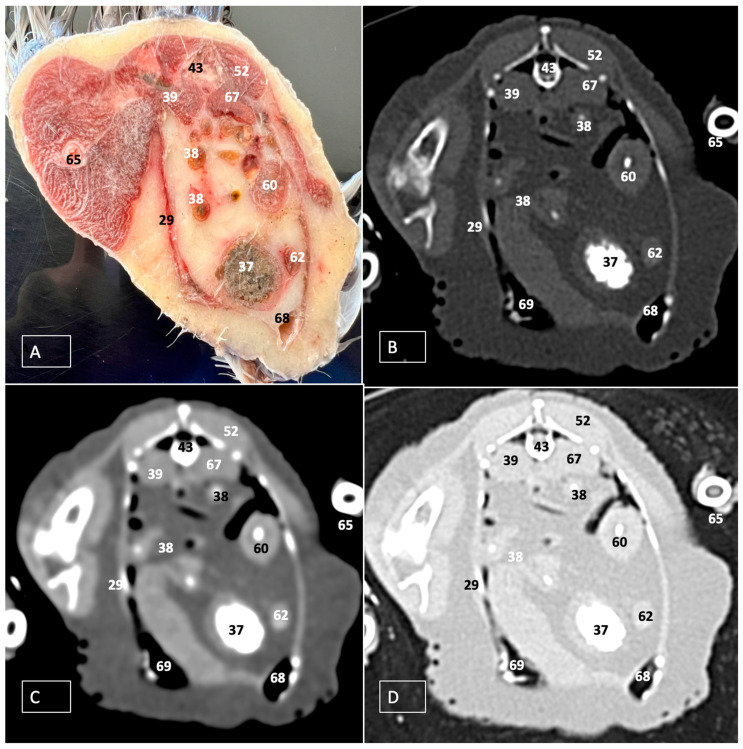
Transverse cross-section (**A**), bone (**B**), soft tissue (**C**), and pulmonary (**D**) CT images of the coelomic cavity of the Cory’s Shearwater at the level of the kidneys, equivalent to line VI in Figure 2. 43: spinal cord. 39: right kidney. 67: left kidney. 38: jejunoileum. 60: proventriculus. 37: ventriculus. 62: duodenum. 69: right abdominal air sac. 68: left abdominal air sac. 29: ribs. 65: femur. 52: musculus levator caudae.

**Figure 9 animals-14-00858-f009:**
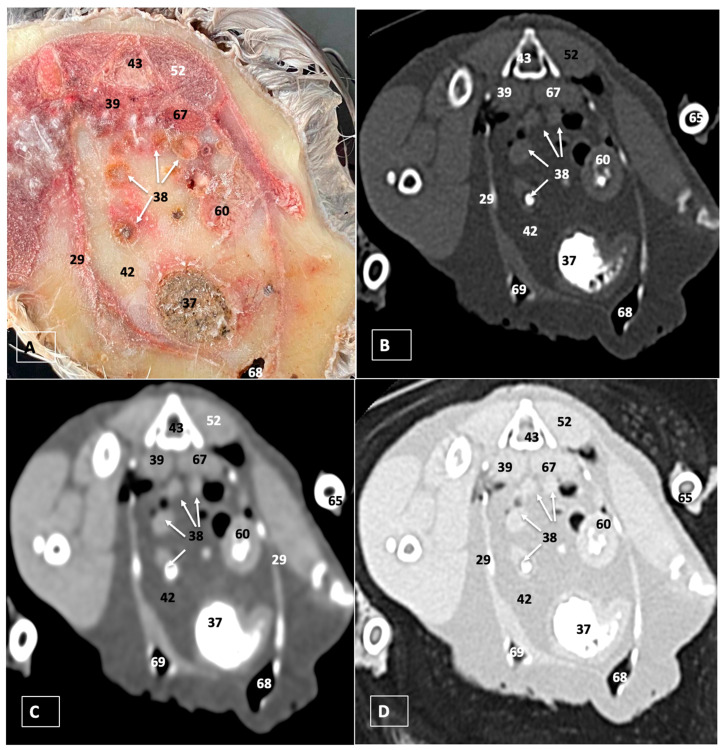
Transverse cross-section (**A**), bone (**B**), soft tissue (**C**), and pulmonary (**D**) CT images of the coelomic cavity of the Cory’s Shearwater at the level of the ventriculus, equivalent to line VII in Figure 2. 43: spinal cord. 39: right kidney. 67: left kidney. 38: jejunoileum. 60: proventriculus. 37: ventriculus. 42: intestinal peritoneal cavity. 69: right abdominal air sac. 68: left abdominal air sac. 29: ribs. 65: femur. 52: musculus levator caudae.

**Figure 10 animals-14-00858-f010:**
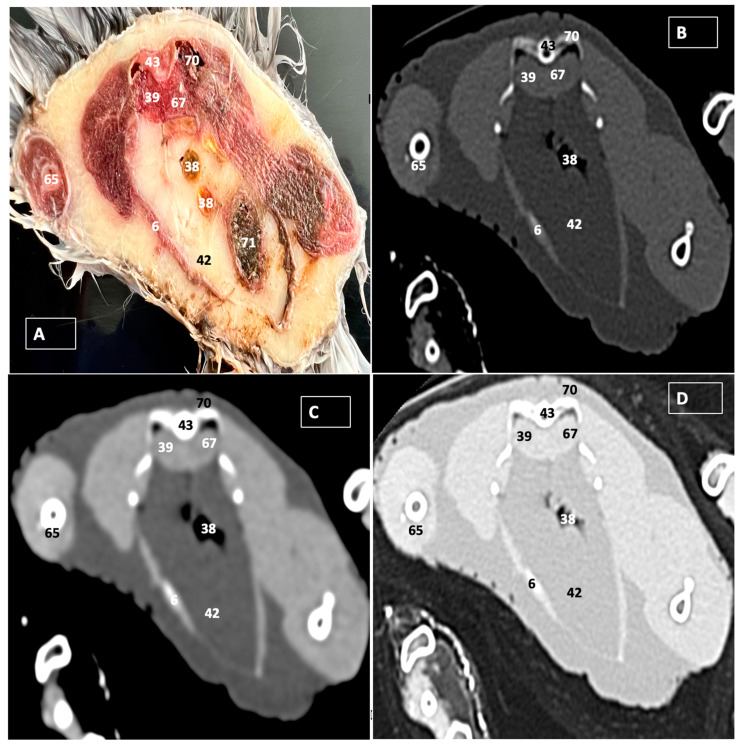
Transverse cross-section (**A**), bone (**B**), soft tissue (**C**), and pulmonary (**D**) CT images of the coelomic cavity of the Cory’s Shearwater at the level of the caecum, equivalent to line VIII in Figure 1. 43: spinal cord. 70: synsacrum. 39: right kidney. 67: left kidney. 42: intestinal peritoneal cavity. 38: jejunoileum. 71: caecum. 65: femur. 6: pectoral muscle (abdominal portion).

**Figure 11 animals-14-00858-f011:**
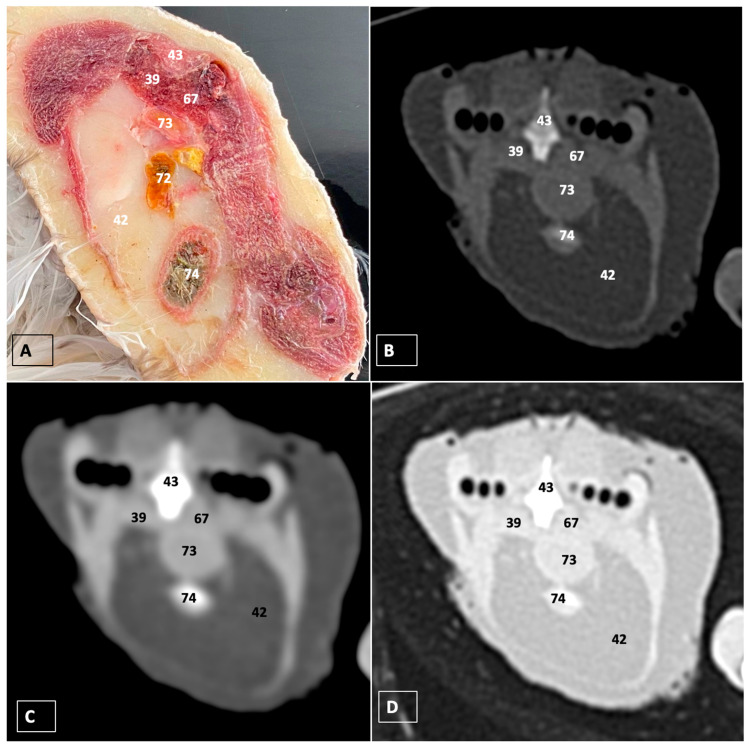
Transverse cross-section (**A**), bone (**B**), soft tissue (**C**), and pulmonary (**D**) CT images of the coelomic cavity of the Cory’s Shearwater at the level of the oviduct and cloaca, equivalent to line IX in Figure 2. 43: spinal cord. 39: right kidney. 67: left kidney. 72: ileum. 42: intestinal peritoneal cavity (full of fat). 73: oviduct. 74: cloaca.

**Figure 12 animals-14-00858-f012:**
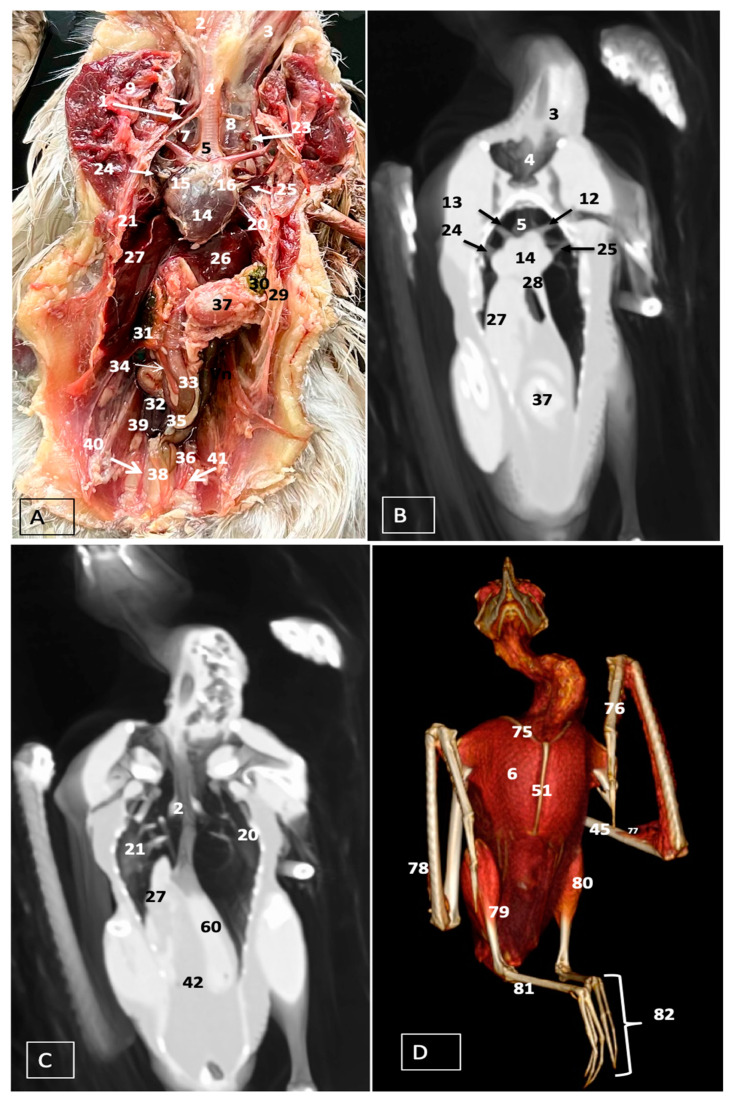
Anatomical dissection illustrating the coelomic cavity of Cory’s Shearwater (**A**) and dorsal MPR volume rendering images in the pulmonary CT window of the coelomic cavity at level of the heart (**B**), the hepatic lobes (**C**), and a 3D CT reconstruction image (**D**). 1: sternotracheal muscle. 21: right lung. 20: left lung. 2: esophagus. 3: longissimus colli muscle. 4: trachea. 5: syrinx. 6: pectoral muscle (thoracobrachialis muscle). 7: right cervical air sac. 8: left cervical air sac. 9: jugular vein. 23: right thyroid. 24: right cranial cava. 25: left cranial cava. 12: left brachiocephalic trunk. 13: right brachiocephalic trunk. 14: heart. 15: right atrium. 16: left atrium. 26: left hepatic lobe. 27: right hepatic lobe. 60: proventriculus. 28: hepatopericardial ligament. 29: ribs: 30: ingesta: 31: spleen. 32: descending duodenum. 33 ascending duodenum. 34: pancreas. 35 duodenal-jejunal flexure. 36: caecum. 37: ventriculus. 38: jejunoileum. 39: right kidney. 40: right ureter. 41: left ureter. 42: intestinal peritoneal cavity. 75: clavicle. 76: metacarpal bone. 77: ventral collateral ligament + cranial cubital ligament. 45: humerus. 51: sternum. 78: ulna. 79: tibiotarsus. 80: fibularis longus muscle. 81: tarsometatarsus. 82: phalanges.

## Data Availability

The data presented in this study are available in this article.

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
