# Peer review of "A Cadaveric Study Using Anatomical Cross-Section and Computed Tomography for the Coelomic Cavity in Juvenile Cory’s Shearwater (Aves, Procellariidae, Calonectris borealis)"

_animals, 2024, doi:10.3390/ani14060858_

Round 1

Reviewer 1 Report

Comments and Suggestions for Authors

The authors describe the anatomy of a marine bird species belonging to Procellariidae family. In particular, they study the coelomatic cavity by means of cross sections and CT methods. The originality of the work is due to the fact that it is a species not yet studied from this point of view and its description can be useful for many professionals who deal with veterinary clinics, biology and ecology of birds.   Some of the authors of this work have already published similar work on another marine species (puffin).  The introduction, material and methods, results and discussion are adequate and well exposed.  The only limitation I can point out is the fact that the observations are conducted on dead animals (corpses) and this could lead to differences with living animals. But after all it is always a limitation of anatomy conducted with dissection. However, the authors could better explain how the birds died and how long and how they were preserved until dissection. This problem could be addressed, or at least reported, in the discussion. Moreover, the birds employed in this study were only juvenile individuals. This datum should be also specified in the title and discussed in the discussion. The references are appropriate.   Thus, I think that this work deserves to be published after these minor points:

  • in the title please specify that the birds were juvenile
  • in the Mat&Met please better specify the origin of the birds, the cause of death, the time between death and dissection. 
  • in the discussion the authors should discuss these limits.

Author Response

Dear Reviewer,

The authors appreciate the comments and suggestion since they have been crucial to improve our manuscript.

We agree about  the important limitation produced by the fact that the observations were conducted on dead animals (corpses), and this could lead to differences with living animals.

Concerning the specific comments:

- As you recommend, We have specified in the title that the birds collected were juvenile specimens.

  • In the Material and Methods, we have added the origin of the birds, the cause of death, and the time between death and dissection.
  • In the discussion section, we have added information concerning  the limits of studies conducted in dead animals.

Reviewer 2 Report

Comments and Suggestions for Authors

This is a very interesting work for the scientific community, since the species of interest is perfect to study many of the unknown aspects of seabirds. It is well laid out, well written, and reads easily. However, there are some aspects regarding the figures that should be improved, as they detract from the quality of the work.

In figures 1B and 1C it is difficult to distinguish anything, because they are small and with too many letters on it. I suggest making both larger and using numbers and arrows.

It is the same concerning Figure 3A, where it is difficult to distinguish the structures that pointed by letters, which are small and with colors that neither stand out very much.

Same comment for figures 4 and 5. In my opinion, too many structures are pointed out, which in both cases makes it very difficult to understand the whole. However, the following figures are better, because the size of the letters is larger and fewer structures are pointed out.

I would totally redo figure 12 because nothing is visible, especially in figure 12A. Again, too many things are pointed out and there is an excess of letters.

Otherwise, I insist that the manuscript is of great interest to the scientific community and that I personally support all this type of initiatives, combining anatomy and imaging.

Comments on the Quality of English Language

No comments

Author Response

Dear Reviewer,

We really appreciate the comments and suggestions since they have been quite helpful to improve the quality of our manuscript.

As you recommend, we have totally redone the figures, making both larger and using numbers and arrows.